# HDG-select: A novel GUI based application for gene selection and classification in high dimensional datasets

**Shilan S. Hameed** [1,2]*, **Rohayanti Hassan**[3], **Wan Haslina Hassan**[1], **Fahmi F. Muhammadsharif**[4], **Liza Abdul Latiff**[5]

**1** Computer Systems and Networks (CSN), Malaysia-Japan International Institute of Technology (MJIIT), Universiti Teknologi Malaysia, Kuala Lumpur, Malaysia, **2** Directorate of Information Technology, Koya University, Koya, Kurdistan Region-F.R., Iraq, **3** School of Computing, Faculty of Engineering, Universiti Teknologi Malaysia, Johor Bahru, Johor, Malaysia, **4** Department of Physics, Faculty of Science and Health, Koya University, Koya, Kurdistan Region-F.R., Iraq, **5** U-BAN Research Group, Razak Faculty of Technology and Informatics, Universiti Teknologi Malaysia, Kuala Lumpur, Malaysia

* shilan.sameen@koyauniversity.org

## Abstract

The selection and classification of genes is essential for the identification of related genes to a specific disease. Developing a user-friendly application with combined statistical rigor and machine learning functionality to help the biomedical researchers and end users is of great importance. In this work, a novel stand-alone application, which is based on graphical user interface (GUI), is developed to perform the full functionality of gene selection and classification in high dimensional datasets. The so-called HDG-select application is validated on eleven high dimensional datasets of the format CSV and GEO soft. The proposed tool uses the efficient algorithm of combined filter-GBPSO-SVM and it was made freely available to users. It was found that the proposed HDG-select outperformed other tools reported in literature and presented a competitive performance, accessibility, and functionality.

## Introduction

The microarray is a tool used to estimate whether mutations in specific genes are present in a particular individual. The most common type of microarray is utilized to measure gene expression, where the expression values of thousands of genes are calculated from the microarray sample [1]. The identification of the most attributed genes to a specific disease can be carried out by means of gene selection and classification of the microarray datasets, wherein various statistical and optimization algorithms are involved. The outcome of accurate selection of attributed genes would ultimately lead to establishing a cost-effective and useful studies on the altered genes [2]. Furthermore, the identified genes help in classifying the clinical samples to normal and disease samples. Gene selection methods are classified into two main types: filter-based methods and wrapper based ones [3, 4]. Filter based methods work separately without using any connected classifier, so they provide the results faster. They are better applied in analyzing high dimensional data of microarray datasets with thousands of genes and hundreds of

**Funding:** RH received a financial support from the Fundamental Research Grant Scheme (FRGS), Ministry of Education and Universiti Teknologi Malaysia under Vote No: RJ130000.7851.5F037.

**Competing interests:** NO authors have competing interests.

samples [5, 6]. The weakness of filter methods is that most of them are unable to establish a useful correlation among the genes and hence there would be the possibility of selecting redundant genes. This drawback acts to reduce the final classifier accuracy if only filter is applied to select the discriminative genes [4]. Hence, the best approach is to use filters in the preliminary selection steps [6]. Wrappers perform better in selecting discriminative genes since they depend on the model hypothesis to train and test in the gene space [4]. However, wrapper-based techniques are heavy and could be a worst choice if they are directly applied on high dimensional datasets without any preprocessing [7].

Many computational methods are failed to extract a small subset of attributed genes in high dimensional datasets because of the presence of various correlations and redundancy among the genes. Interestingly, studies in the field of cancer informatics have shown a splendid contribution of data mining and machine learning to find the attributed genes [8–11]. It is however proved that machine learning can perform well in cancer classification, it is yet required further improvement and robustness in terms of efficiency and computational cost, especially when high dimensional datasets are investigated. This is because high dimensional datasets contain several redundant and variant genes expression, which in turn acts upon reducing the accuracy and efficiency of the computational techniques used to mine the most attributed genes [12]. In general, the noise in gene expression level is occurred due to biological variations associated with the experiments or the existence of alterations in the genes [4, 13]. Therefore, it is not an easy or a straightforward task to find the attributed genes in high dimensional datasets unless a careful analysis and selection rule is carried out.

Along this line, a binary variant of Harris hawk's optimizer (HHO) was proposed to boost the efficacy of wrapper-based gene selection in high dimensional dataset [14]. Besides, a two-stage sparse logistic regression was reported aiming at obtaining efficient subset of genes with high classification capabilities [15]. That is by combining the screening approach as filter method and adaptive lasso with a new weight as wrapper method. Gene selection in high-dimensional colon cancer microarray dataset was seen to be enhanced by using an ensemble of gene selection technique based on $t$-test and GA [16]. After preprocessing the data using $t$-test, a Nested-GA was employed to get the optimal subset of genes. As such, various approaches were reported in literature in order to increase the gene selection efficacy in high dimensional datasets such as hybrid binary coral reefs optimization algorithm with simulated annealing [17], ensembles of regularized regression models with resampling-based lasso [18], variable-size cooperative coevolutionary particle swarm optimization [19], hybrid dimensionality reduction forest with pruning [20], hybrid feature selection based on reliefF and binary dragonfly [21] as well as hybrid rough set theory and hypergraph [22]. It is observed that the effective approach for gene selection in microarray dataset can be a combination of filter and wrapper algorithms. Obviously, there exist numerous techniques used to select attributed genes in high dimensional datasets, however the complexity of the algorithms and computational cost are limiting their reproducibility with rapid selection of discriminated genes in massive datasets. Nevertheless, particle swarm optimization (PSO), as a searching strategy for genes selection, is proved to be more efficient and easy to implement compared to other methods [23, 24]. This is because few parameters are needed to perform its adjustment and therefore it saves memory. The modified geometric binary of PSO (GBPSO) was effectively utilized for gene selection in autism dataset [12]. Details on PSO and its GBPSO variant can be found in literature [12, 24–27]. GBPSO can be used as a wrapper feature selection method with a support vector machine (SVM). SVMs represent a group of supervised machine-learning methods which were developed by Vapnik [28]. The various forms of this algorithm are widely used [9, 24, 29], particularly for medical related data classification [30–34]. Moreover, SVM can perform both linear and nonlinear separable data classification. When using SVM, it is essential

that the number of coefficients to be determined are primarily based on the number of samples not on the number of genes. In the case of gene classification, SVM utilizes kernel functions to get an orthogonal hyperplane to separate the genes in a specific dimension. Different types of kernels can be applied [24, 35, 36], whereas each kernel type is appropriate for different data. In the current work, a polynomial kernel was utilized for the SVM due to its highest classification accuracy when it is applied for high dimensional datasets.

It is well-known that the process of gene selection and classification is becoming tedious and time consuming when the datasets are not curated such as soft GEO datasets. A review of literature showed that there are various tools created for sequence and genomic data analysis [37–40], while there has been few applications established for gene selection and classification [41–43]. For instance, a java GUI application was developed for microarray data classification using SVM classifier [43]. The researchers concluded that the application performs well when a radial basis SVM kernel is used. However, their tool is not accessible now and it is created only for classification. The varSelRF package and GeneSrF tool were developed for gene selection given the associated error of classification using R language and python [41]. The package of varSelRF can be only used on Linux and Unix OS, while GeneSrF is a web-based tool and is not currently accessible. In another study [42], ArrayMining.net, which is a web-based tool, was constructed for gene selection and class identification using supervised and un-supervised techniques. In the current work, a novel user-friendly and stand-alone (non-web based) application is proposed for a simple and efficient gene selection and classification in the high dimensional datasets. The software program was developed with the help of interfacing MATLAB with Weka tool, combining their benefits in one package. The proposed application is named as HDG-select, referring to its capability of high dimensional gene selection. It can be used by researchers and students to reduce the burden of hard-working steps of dataset curation, gene selection and classification on a one-platform scheme. The main advantages of the proposed application include dataset curation, user-defined gene filtration, handling both numerical and categorical samples and combining the functionality of MATLAB and Weka in a single tool. If someone wants to perform a complete gene selection including dataset curation and filtration, a comprehensive coding is first required in MATLAB and then the results need to be transferred to Weka tool in order to run the GBPSO-SVM algorithm. Interestingly, the proposed HDG-select is the collection of all necessary operations within a single user-friendly graphical interface, which helps the users to practice simplicity, accuracy and reduced computational cost. Noticeably, the tool uses a combination of filters and GBPSO wrapper for gene selection, while SVM is used for classification. Furthermore, the reported tools in literature accept CSV files as input datasets. However, our developed tool can handle both CSV and.Soft file formats, which is specifically useful for analysing the non-curated genomic data available in the GEO database.

## Materials and methods

### Implementation procedure

The process of selection and classification of genes in high dimensional microarrays using the developed HDG-select tool and the built-in structure of the application are shown in Figs 1 and 2, respectively. The first step was to reduce the dimensionality of the datasets by removing the redundant/irrelevant genes whose expression values are close amid the control and non-control classes. For this purpose, the values of mean and median ratio were calculated based on the variance of the genes expression, which is discussed later in detail. This process is performed so that the next steps become more efficient and easier. Later on, two different filters, namely *t*-test (TT) and Wilcoxon rank sum (WRS), are used to filter the desired number of top

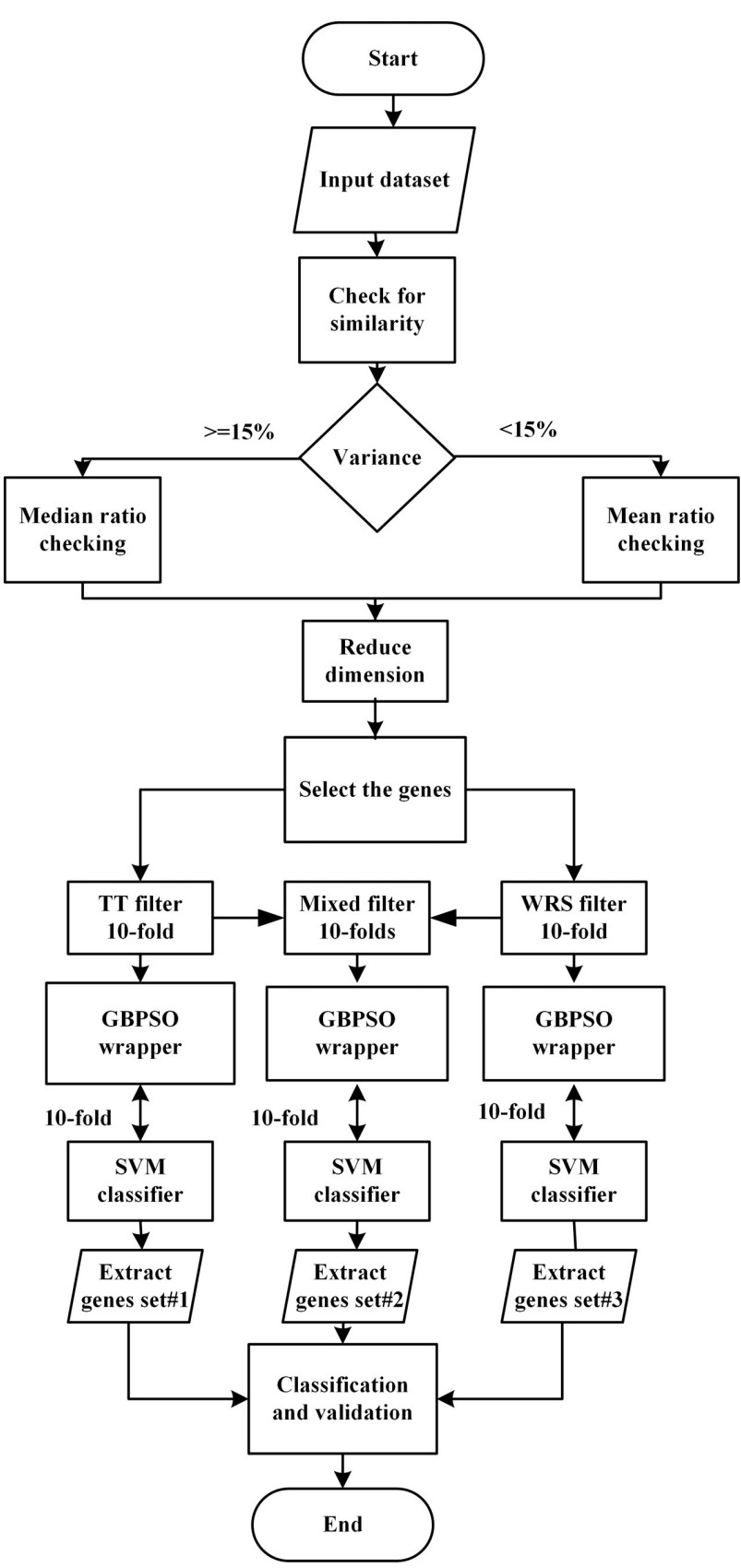

**Fig 1. The flowchart of selecting genes and classification in high dimensional datasets using the HDG-select application.** The highly irrelevant genes are fist removed (dataset curation) by considering the values of mean and median ratio, followed by the use of different filters in combination with the GBPSO algorithm.

relevant genes. These filters and their combination process are elaborated in the next sections. As shown in Fig 2, the curation and filtration steps are implemented by MATLAB coding, while the use of wrapper based GBPSO-SVM algorithm is realized by Java programming, that is by interfacing the GBPSO-SVM algorithm from Weka with MATLAB.

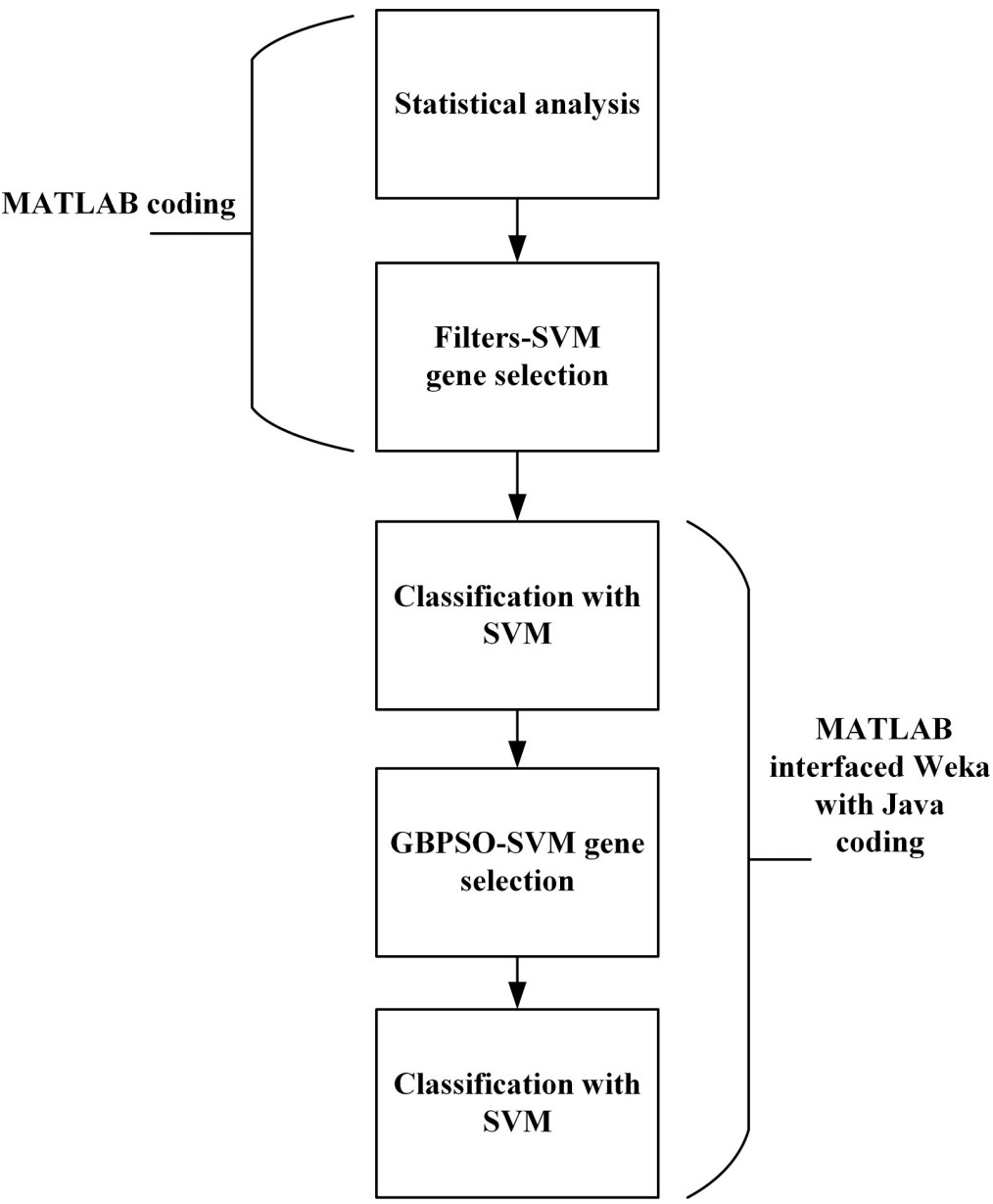

**Fig 2. The built-in structure of the developed HDG-select application.** The curation and filtration steps are implemented by MATLAB coding, while the use of wrapper based GBPSO-SVM algorithm is realized by MATLAB interfaced Weka through Java coding.

## Microarray datasets

Validation and assessment of the developed HDG-select application was carried out by testing 11 high dimensional datasets of different types of diseases. The characteristics of the datasets are given in Tables 1 and 2. The first six datasets in Table 1 are in csv format, which were pre-processed and previously used for gene expression analysis [11]. The Leukemia cancer dataset was achieved from [44]. The dataset of leukemia cancer is given as S1 Dataset. The colon cancer microarray dataset was originally analyzed by Alon *et al* [45], it is given as S2 Dataset. The prostate cancer dataset is based on oligonucleotide microarray, which was obtained from [46]. The dataset of prostate cancer is given as S3 Dataset. The rest of the datasets (Breast, CNS and Ovarian) were achieved from [47] whose datasets are given as S4–S6 Datasets, respectively.

The second batch of investigated datasets are in soft format with originally non-curated condition, which can be downloaded from the well-known public repository of GEO (NCBI) [48] under GDS file name. The main characteristics of these datasets are given in Table 2 with some descriptions as follows:

**Inflammatory breast cancer (GDS3097).** Tumor epithelium and underlying stromal cells were extracted using laser capture microdissection of human breast cancer to study gene expression variations based on inflammatory and non-inflammatory breast cancer tissue types.

**Breast cancer (GDS3716).** With Affymetrix HU133A microarrays, 42 total laser capture micro dissected histologically normal samples of breast tissue were analyzed in this dataset.

**Brain metastatic breast cancer (GDS5306).** Gene expression of 19 HER2+ breast cancer brain metastases were comparably examined with HER2+ nonmetastatic primary tumors.

**Autism disorder (GDS4431).** Total RNA was extracted for microarray experiments with Affymetrix Human U133 Plus 2.0 39 Expression Arrays. The autistic samples were diagnosed by medical professionals of developmental pediatrician and psychologist according to the DSM-IV criteria and the diagnosis was confirmed on the basis of ADOS and ADI-R criteria [49].

**Influenza A (GDS6063).** More than 2600 genes were expressed differently in pDCs exposed to influenza A compared to controls (no viruses) blood pDCs.

## Dimensionality reduction of soft format datasets using mean and median ratio

Because of the presence of variance among the genes expression in high dimensional datasets and the non-curated nature of GEO soft datasets [48, 50], it is imperative to perform a pre-processing mechanism in order to reduce the dimensionality of the datasets, thereby removing the redundant and highly irrelevant genes. For this purpose, the overall similarity of genes expression was assessed through the estimation of their mean and median values among the two

**Table 1. The main characteristics of the pre-reduced high dimensional datasets in csv format.**

| Dataset | #Genes | #Samples | #Class (class1:class2) |
|---|---|---|---|
| Leukemia | 3051 | 72 | 2(25:47) |
| Colon | 2000 | 62 | 2(22:40) |
| Prostate | 6033 | 102 | 2(50:52) |
| Breast | 24481 | 97 | 2(46:51) |
| CNS[a] | 7129 | 72 | 2(21:39) |
| Ovarian | 15154 | 253 | 2(162:91) |

[a] Central Nervous system.

**Table 2. The main characteristics of the original/non-curated GEO datasets in soft format.**

| Dataset | #Genes | #Samples | #Class (class1:class2) |
|---|---|---|---|
| Inflammatory Breast Cancer (GDS3097) | 22283 | 48 | 2(35 NIBC:13 IBC) |
| Breast Cancer (GDS3716) | 22283 | 42 | 2(24 control:18 breast cancer) |
| Brain Metastatic Breast Cancer (GDS5306) | 61359 | 38 | 2(19(BMBC) tumor:19(NBC) tumor) |
| Autism (GDS4431) | 54675 | 146 | 2(69 control:77 autism) |
| Influenza A (GDS6063) | 48107 | 10 | 2(5 postive:5negative) |

classes. When a mean criterion is applied to identify the redundant/irrelevant genes, the mean of genes expression is determined. Similarly, when the median criterion is considered, the median value of genes expressions is calculated.

It was seen that when there is a high variance in the genes expression (variance $\geq$ 15%), the application of median criterion to reduce the dataset dimensionality is performed better in comparison to the application of mean criterion [12]. This is because the mean value of genes expression is affected by the high variance. Therefore, in this work, median criterion is applied on the genes of variance $\geq$ 15%, while mean criterion is used for those with variance $<$ 15%. Consequently, genes whose median and mean ratio of their expression are between 0.95 and 1/0.95 are removed from the dataset. This threshold range is chosen intentionally in order to remove the redundant and less significant genes from the whole dataset, and hence making the next steps of the gene selection simple and cost-effective without compromising the selection accuracy.

## Gene selection using statistical filters

The second step of gene selection in the GEO soft datasets, after the dimensionality reduction, was performed by using two different statistical filters and their combination, namely two-sample t-test (TT), Wilcoxon rank sum test (WRS) and combined TT-WRS. However, this selection by means of the filters can be the first step of gene selection for the pre-curated data-sets of CSV format because the proposed HDG-select tool allows users to bypass the dimensionality reduction step for the pre-curated CSV datasets. The choice of these filters is based on the findings that the TT and WRS filters performed well when they used for gene selection in the high dimensional dataset of autism [12]. This is because each filter is based on different assumptions related to the mean, median and variance which can be found in the high dimensional datasets. Because the filtration power is different for each filter, the combination of them might yield a better selection performance [11, 51, 52]. The TT filter was applied to microarray genes [53] and it was seen that the filter shows a strong scalability when the number of genes is high [54]. Hence, some researchers have used the TT filter as the only step of gene selection [55, 56]. Also, WRS filter was effectively used for the pre-selection of genes [57, 58], especially when the data are associated with high variance [59].

In this work, the statistical filtrations are applied on the datasets in a 10-fold run in order to avoid overfitting. As such, the genes are ranked among the 10-fold from the most significant to the least significant ones. Hence, based on their ranking position (weight), the desired number of most highly ranked genes can be extracted. The equation used to weigh the genes and ranking their positions based on their significance is a formula of global weight that is given by:

$$w(f) = \sum_{i=1}^{K} w_i(f) \tag{1}$$

where each $i$ in $K$ = the number of current fold iterations in the entire 10-fold run.

The *t*-test (TT) filter is a univariate filter that is commonly used for binary classes [53, 54]. The general assumption of the *t*-test is that the values are uniformly distributed with a bell-shaped distribution curve among the two classes. The *t*-test null hypothesis supposes equal means and equal variances, and this assertion is rejected by the alternative hypothesis. The *t*—test formula is [60]:

$$t = \frac{c_1 - c_2}{\sqrt{\frac{\sigma_1^2}{n} + \frac{\sigma_2^2}{m}}} \tag{2}$$

where *n* and *m* are the first- and second-class population size, respectively. The result of the evaluation calls *t*, its value is ranged from 0 to 1 based on the significancy level. The value of 1 refers the abandonment of the null hypothesis at the 5 percent and 0 refers to the acceptance of the null hypothesis at the same level of significance. The test also returns the probability value of *t*. The lower *p*-value implies a noticeable difference between the compared samples. The parametric form of TT filter assumes equal variance and normal distribution, while non-parametric one assumes unequal variance and random distribution. In this work, the non-parametric TT filter is used because most of the data distribution in high dimensional datasets follow unequal variance and random distribution due to the presence of high noise and various expression values.

The second filter is Wilcoxon rank sum (WRS) test, which is a non-parametric filter method [61]. Hence, it is not essential for the gene values in the classes to have a normal distribution such as seen in the high dimensional datasets. This method is also known as the Mann-Whitney test [62, 63]. It uses a median based criterion to distinguish between the two classes. The test compares the samples medians and provides results on a ranking manner rather than in numerical values [64]. The index value and rank for each element in the result can be determined by arranging them in an ascending order. The null hypothesis considered by WRS test is that all genes originate from one class. The statistical formula of the Wilcoxon rank sum is as follows [57]:

$$s(g) = \sum_{i \in N_0} \sum_{j \in N_1} I\left(\left(\mathbf{x_j^{(g)}} - \mathbf{x_i^{(g)}}\right)\right) \leq 0 \tag{3}$$

where *I* is the function used to distinguish the classes. If the logical expression $\left(\mathbf{x_j^{(g)}} - \mathbf{x_i^{(g)}}\right) \leq 0$ is true, *I* is 1; otherwise, it is 0. $\mathbf{x_i^{(g)}}$ is the expression value of gene *g* in sample *I*, $N_0$ and $N_1$ represent the number of observations in each of the two classes, respectively, and *s(g)* denotes the difference in the expression of the gene in the two classes. Based on whether *s(g)* becomes 0 or reaches the maximum of $N_0 \times N_1$, the considered gene is ranked in importance in the classification process. The following equation is used to calculate the gene's importance:

$$q(g) = \max\left(s(g), N_o \times N_1 - s(g)\right) \tag{4}$$

### Gene selection using GBPSO-SVM algorithm

In the final step of gene selection, the wrapper based GBPSO-SVM algorithm is applied. The GBPSO uses SVM's accuracy prediction to select the best subset of genes. SVM algorithm was used with GBPSO due to its sufficient ability in giving sensible classification accuracy for microarray data regardless of the number of samples. This is a useful feature of SVM for microarray data due to the low sample-to-gene ratio in this dataset. GBPSO starts with a number of randomly selected genes, then in each iteration it searches for the optimum subset of genes. The SVM classifier assesses the performance of each candidate subset using 10-fold cross validation. Hence, every current candidate subset of genes is commonly better than the

previous subset. The GBPSO original package of the algorithm can be retrieved from [65]. In the current work, a polynomial kernel was utilized for the SVM due to its highest classification accuracy when it is applied for high dimensional datasets.

## Development of HDG-select application

The so-called HDG-select application was created using graphical user interface (GUI) in MATLAB. The developed application has a user-friendly interface which is easy to understand and implement for gene selection and classification in both of high dimensional and normal datasets. The first two steps of gene selection were written in MATLAB, while the third step was written in Java, taking the advantage of Weka packages and Java interfacing [66]. Meanwhile, we used Java coding for interfacing the Weka functionality with MATLAB. It was designed in a way that it can handle errors and control the user's inputs to perform each step correctly. This was achieved by using message handlers during the application process. The HDG-select application was made freely available to users, which can be downloaded from GitHub (https://github.com/Shilan-Jaff/HDG_select). Fig 3 shows the interface of the tool, which is composed of four major sections described as below:

a. *Input (Dataset import)*: The user is able to import two formats of datasets, namely soft and csv, as shown in part (a) of Fig 7. Most of the curated datasets available online are in the form of csv format, which is the common format for machine learning applications. The other dataset type is soft file, which is the format of the gene expression profiling datasets made available to public at GEO NCBI database [48]. This type of dataset is usually in the form of a non-curated and high dimensional structure. Before importing the csv files, users have to make sure that the last column contains the class label, while for the soft file the range of the sample class should be manually given to the application. This is because information regarding the sample class is inherently not presented in the soft files, so it must be obtained from the dataset description given in the NCBI database.

b. *Preprocessing*: In this step of analysis, a reduction process can be made upon the high dimensional datasets, or upon the datasets that have not yet been reduced/curated by researchers. As such, the dataset will be easily handled for the next steps of analysis. It can be seen from part (b) of Fig 7 that this section has two options, which allows the user to choose between reducing the dataset or leaving it as it is. However, for the soft format

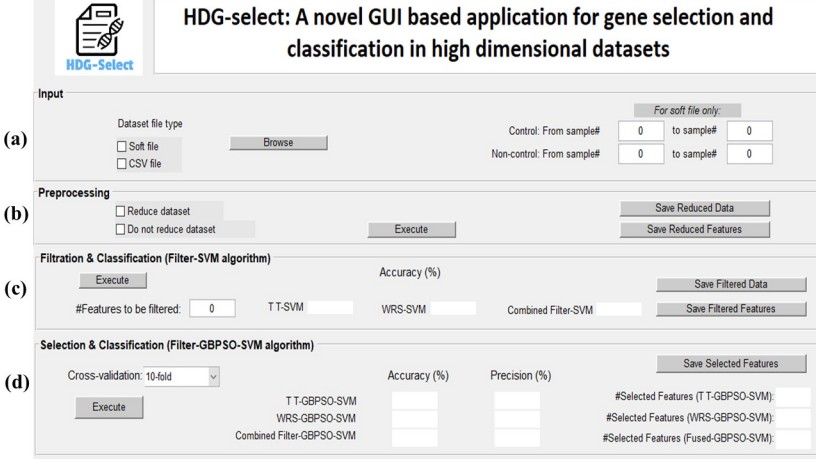

**Fig 3. The main interface of the developed HDG-select application used for gene selection and classification in high dimensional datasets.**

datasets, gene reduction is obligatory, otherwise the process would be computationally costly and memory overload is resulted. At the end of this process, users can save the reduced dataset for their future use if required.

c. *Filtration and classification*: Once the user finalized the preprocessing step, the dataset is proceeded to the next stage of filtering the most significant genes, classification assessment and saving the filtered dataset, as shown in part (c) of Fig 7. It is worth to mention that with the help of this application, the user can get accessibility to decide on the number of genes to be filtered. Hence, one can choose the optimum filtered genes based on the preference and understanding of the dataset. Nevertheless, the default number of gene filtration was set to be 200.

d. *Selection and classification using Filter-GBPSO-SVM algorithm*: The last and most important action is to select the genes and to apply the SVM classifier on the selected genes, as shown in part (d) of Fig 7. This is applied on the results achieved from previous steps and is performed on each dataset generated from the filtration step. Here, the user can see how many genes are selected by each approach and has access to save them.

The HDG-select application uses the following equations to determine the accuracy and precision of the gene selection and classification, respectively.

$$Accuracy = \frac{TP + TN}{TP + TN + FN + FP} \tag{5}$$

$$Precision = \frac{TP}{TP + FP} \tag{6}$$

Where *TP*, *TN*, *FN*, *FP* are the true positive, true negative, false negative and false positive detected samples, respectively.

## Results and discussions

In order to show the functionality and robustness of the proposed HDG-select tool, a step by step analysis is presented. As we mentioned earlier, users can choose to not perform the dimensionality reduction on the pre-curated CSV datasets. However, the preprocessing step (see Fig 3B) for the soft GEO datasets is a must since these datasets are not pre-curated. It leads to a heavy computational burden and low classification accuracy if they are directly applied for gene selection. Consequently, the soft GEO files were preprocessed and the dataset dimensionality was interestingly reduced. For example, the genes in Autism and influenza A datasets were reduced from 54613 to 14530 and from 22283 to 17939, respectively upon the application of the preprocessing step. As such, with the help of HDG-select, the impact of filtered genes on the SVM classification accuracy in the soft GEO datasets was investigated, as shown in Fig 4. Results showed that limiting the filtered genes to below 150 genes has negatively affected the classification accuracy, except for the influenza A dataset which showed a stable performance regardless of the change in the genes number. Concludingly, the dataset with a steady curve indicating the presence of good correlation between the genes and hence extra filtration does not further improve the classification accuracy. Nevertheless, filtering the genes to a low possible number can save memory and speed up the execution time in the final stage of gene selection and classification.

Fig 5A and 5B show the SVM accuracy of gene selection in CSV datasets and soft GEO datasets after the application of TT filter and TT filter-GBPSO algorithm with the help of the

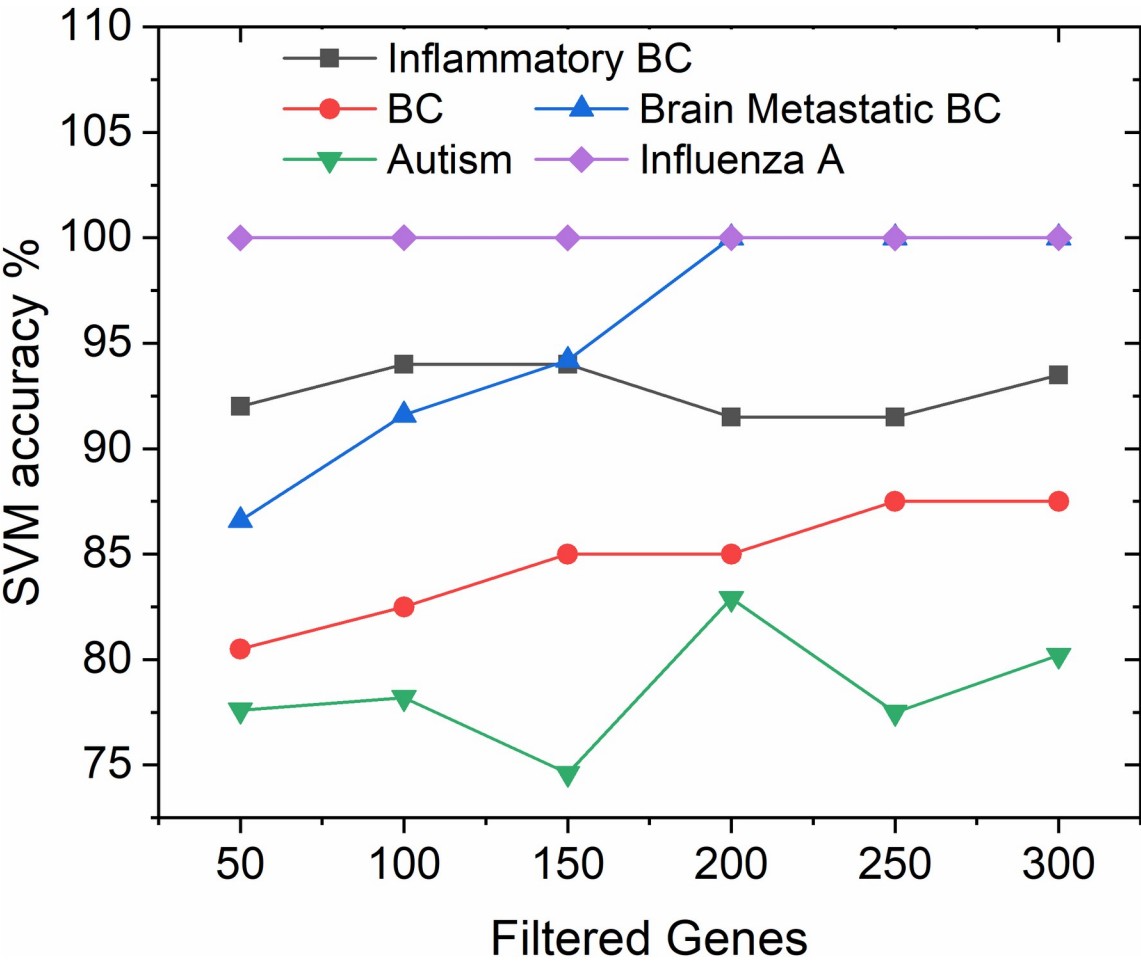

**Fig 4. Effect of the number of filtered genes on the SVM classification accuracy in the first step of gene selection in high dimensional datasets.**

proposed HDG-select, respectively. More data results from the application of different filters and GBPSO are given as S1 and S2 Tables. One can notice from the results that the accuracy of SVM is largely improved when the statistical filters are used in combination with GBPSO. The application of filters has improved the classification accuracy when it is compared to the results of the original dataset. Comparably, the use of GBPSO algorithm in combination with the filters has led to improved performance. For instance, the classification accuracy in leukemia and colon cancer has reached 100% when a combined TT-WRS filter with GBPSO was utilized, surpassing the results obtained by Nested-GA algorithm [16]. Noteworthy, the SVM accuracy for brain metastatic BC and Influenza A datasets remained 100% in both steps of genes selection by filters and GBPSO-SVM, indicating that the HDG-select tool has a strong power on the dimensionality reduction of the datasets to maintain the most important genes that are quite useful for the subsequent steps of gene selection process.

Results showed that the best approach to increase the accuracy of classification of the gene selection in high dimensional datasets is to utilize different filters in combination with the GBPSO-SVM algorithm, as shown in Fig 6. It was observed that the combination of TT-WRS filter with GBPSO has led to improve the SVM accuracy in eight datasets out of eleven datasets.

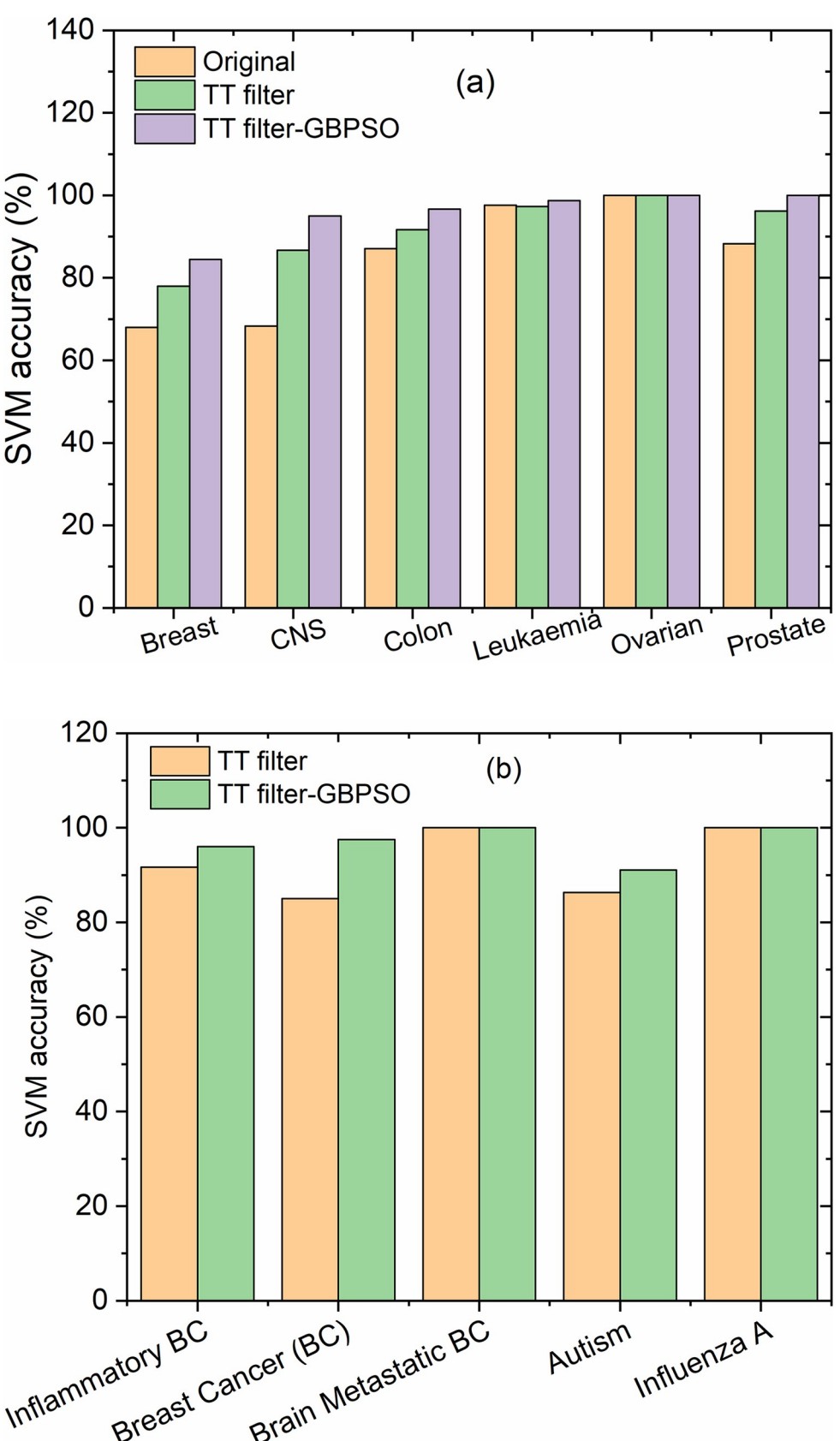

**Fig 5.** A representative comparison of SVM accuracy of gene selection in the CSV datasets (a) and soft GEO datasets (b) after the application of TT filter and TT filter-GBPSO algorithm using the HDG-select application.

It is worth to mention that the use of HDG-select tool is also important even if the accuracy is not much improved after the selection process because the HDG-select toll can help in selecting a small subset of the attributed genes while maintaining the original accuracy but reducing the computational burden.

Fig 7 shows the achieved accuracy and precision of gene selection from various high dimensional datasets using the proposed HDG-select application. It was seen from the results that the values of accuracy and precision are in the range from 90% to 100% for different datasets. For instance, the precision of gene selection in the Prostate, Ovarian, Inflammatory BC and Influenza A datasets has reached 100% which is close enough to their classification accuracy. Hence, it can be concluded from the coincidence of the accuracy and precision data that the

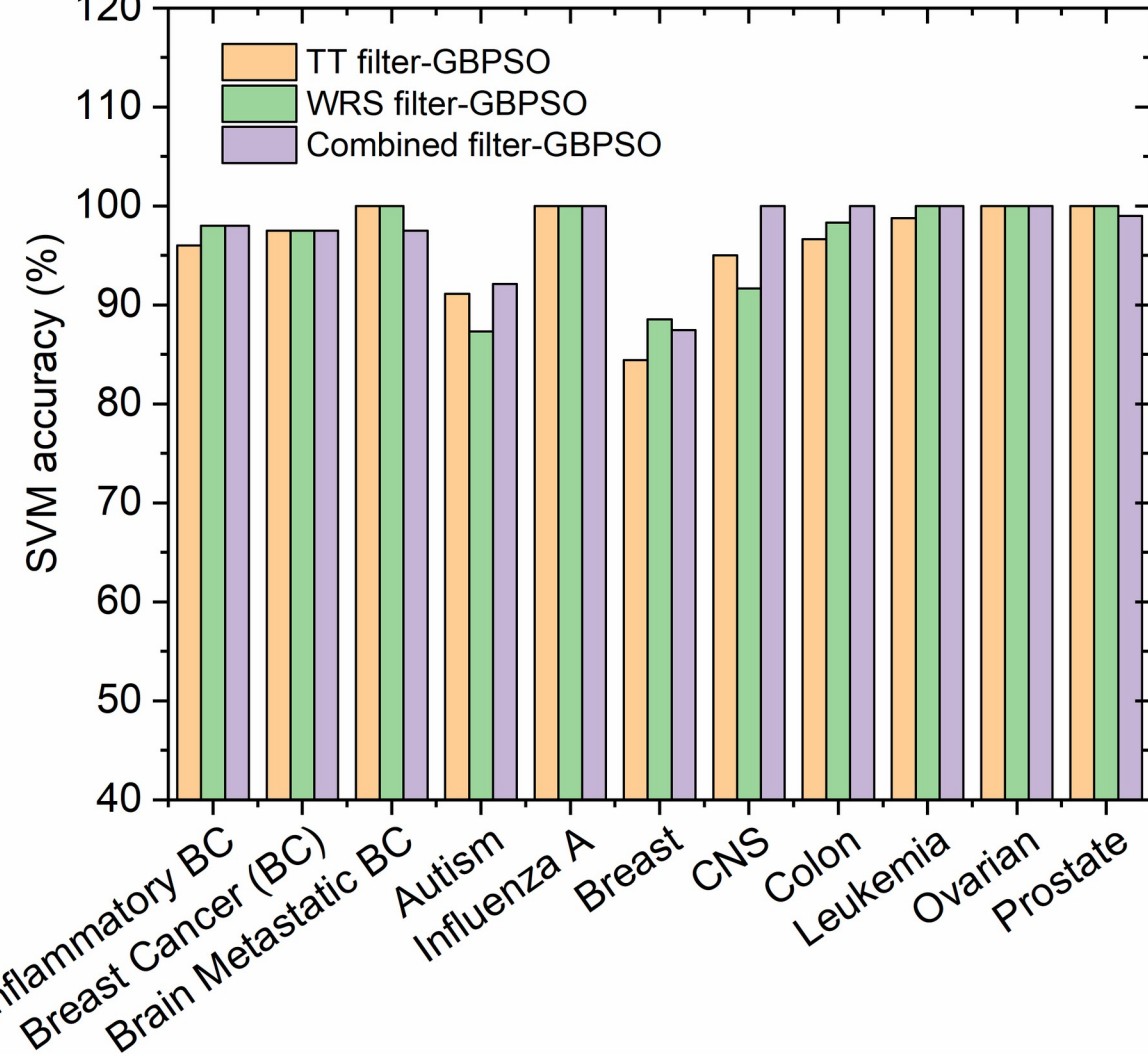

**Fig 6. Comparison of the SVM accuracy in selected genes by different filters-GBPSO-SVM approach using the proposed HDG-select application.**

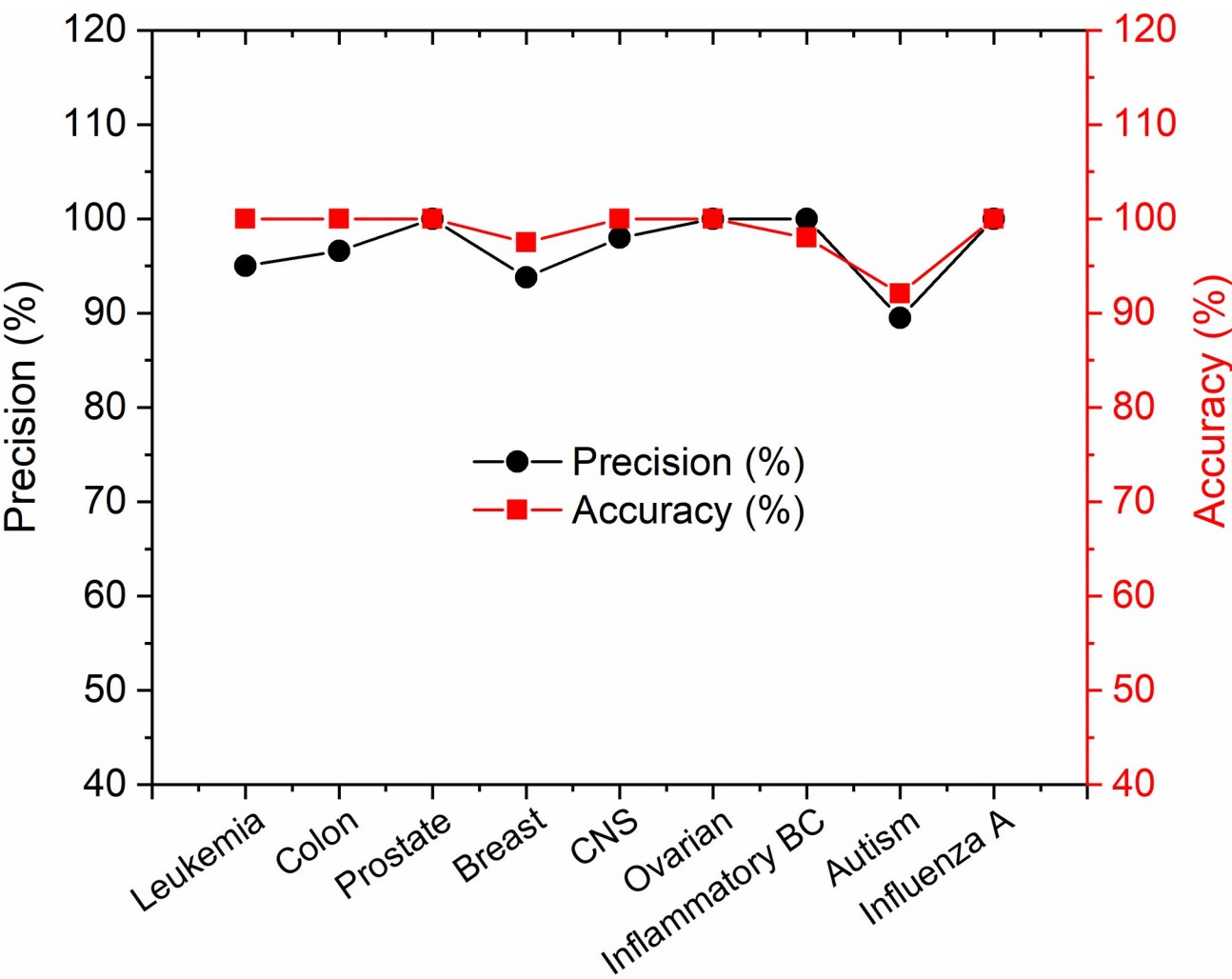

**Fig 7. Accuracy and precision of the classification of datasets after final gene selection using the proposed HDG-select application.**

proposed HDG-select has performed very well on various datasets when it was used to select the most attributed genes efficiently, thereby providing a competitive classification accuracy.

Furthermore, a comparison of the results obtained by the HDG-select application to those reported in literature suggesting that the filter-GBPSO-SVM algorithm is more efficient than PCC-BPSO-SVM and PCC-GA-SVM algorithm [11] when it comes to the selection and classification of attributed genes in high dimensional datasets, as shown in Fig 8 and S3 Table.

In order to show the effectiveness of the proposed HDG-select application in the final step of selecting the biomarker/attributed genes, heatmap graphical analysis was plotted. Figs 9 and 10 show the produced heatmap of seven biomarker genes versus the samples for breast cancer and influenza A datasets. It can be seen from the heatmap that the biomarker genes show a high discrimination ability between the control and non-control samples. For instance, gene number 1 and 3 have the highest discrimination ability among the breast cancer and influenza A datasets, respectively.

Table 3 shows a detailed comparison of our proposed application with those reported in literature. It was concluded that the proposed HDG-select outperformed the other tools in terms of overall performance, accessibility and functionality. Noticeably, the most competitive tool

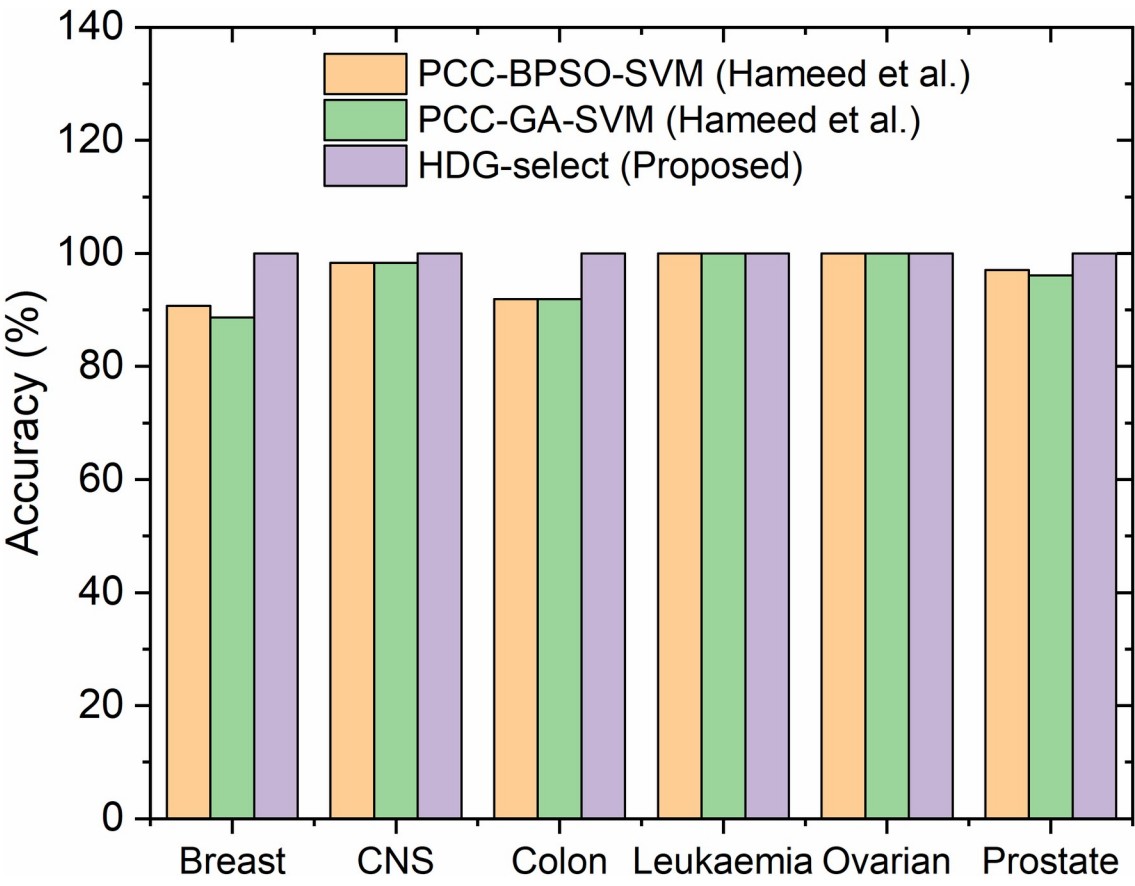

**Fig 8. Comparison of the classification accuracy in genes selection by different methods.**

to the proposed HDG-select can be ArrayMining [42]. However, this tool accepts dataset files of CSV format only, while with the help of HDG-select one can also perform dataset curation and dimensionality reduction on the soft GEO datasets. Furthermore, with the proposed HDG-select a multiple gene selection and classification can be performed simultaneously and the selected genes with their expression can be downloaded in CSV format, while ArrayMining [42] can perform one task at each time and the selected genes is downloadable in text format. Table 4 shows a comparison between HDG-select and ArrayMining tool for two representative CSV datasets that were previously analyzed by ArrayMining in terms of accuracy and precision.

## Conclusions

A novel GUI based stand-alone application, named as HDG-select, was developed to select and classify the attributed genes in high dimensional datasets effectively. The application was validated on 11 datasets and it was found to perform well on most of high dimensional data-sets, including CSV and GEO soft file formats. The proposed HDG-select tool uses efficient algorithm of combined filter-GBPSO-SVM. It was observed that the best approach of increasing gene selection efficiency in high dimensional data is to utilize a mixed filter-GBPSO-SVM algorithm. It was concluded that the proposed HDG-select outperformed the other tools in terms of overall performance, accessibility, and functionality.

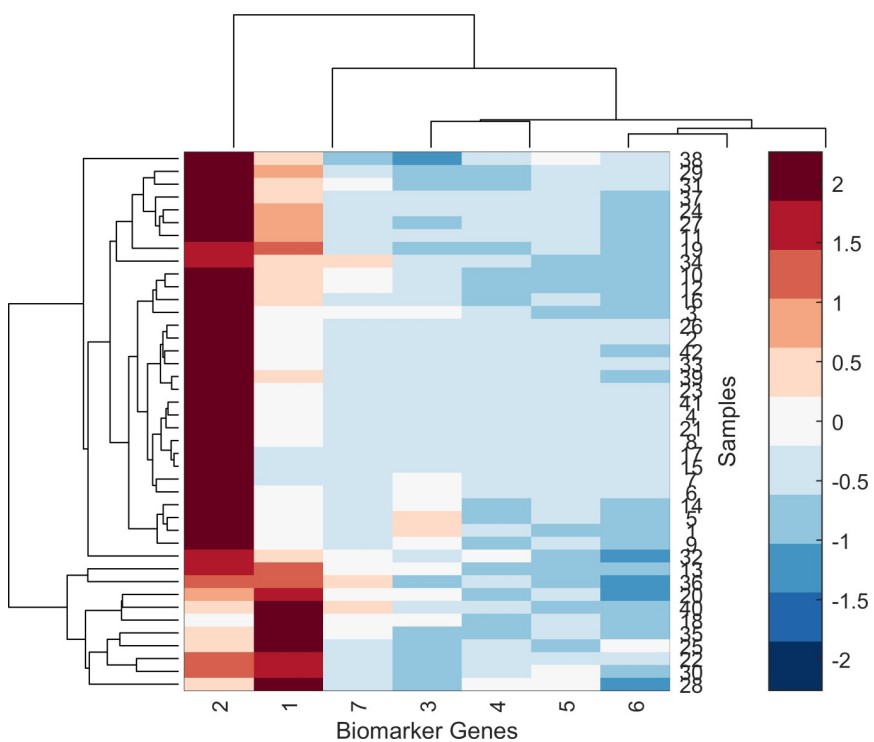

**Fig 9. The heatmap of seven selected biomarker genes of breast cancer using the proposed HDG-select application.**

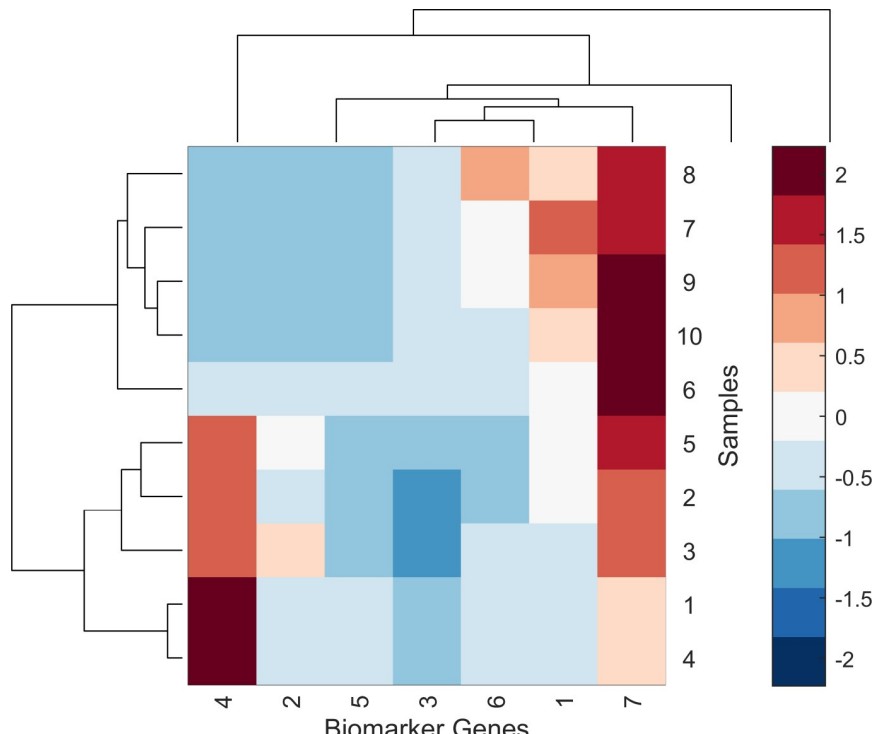

**Fig 10. The heatmap of seven selected biomarker genes of influenza A using the proposed HDG-select application.**

**Table 3. Comparison of the proposed HDG-select tool with those reported in literature for gene selection and classification.**

| Application name | language/ package | Gene selection/ classification | Algorithm | Accessibility (online/offline) | GUI Interface | Operating system | Dataset format | User-friendliness |
|---|---|---|---|---|---|---|---|---|
| SVM Classifier [43] | Java | No/Yes | SVM | No (online) | Yes | No-restriction | Not known | Low |
| R. GeneSrF and varSelRF [41] | R, Python | Yes/No | Random forest | No (online) | No | Linux, Unix for R package | CSV | Low |
| ArrayMining [42] | R, C++ and a PHP-interface | Yes/Yes | Filter Classification clustering | Yes (online) | Yes | No restriction | CSV | Medium |
| HDG-select (proposed) | Weka, Java and MATLAB | Yes/Yes | Two filters, their combination, GBPSO wrapper and SVM classifier | Yes (offline-no internet required) | Yes | No restriction | CSV and (. soft) GEO dataset | High |

**Table 4. Comparison of the proposed HDG-select tool with ArrayMining tool in terms of accuracy and precision of gene selection and classification in high dimensional datasets.**

| Dataset | Application name | Algorithm (# selected genes) | SVM classification accuracy (%) | Precision (%) |
|---|---|---|---|---|
| Colon | ArrayMining [42] | Filter (80) | 80.7±13 | 86.6 |
| | HDG-select (Proposed) | Filter-wrapper (30) | 98.3±1.7 | 96.6 |
| Prostate | ArrayMining [42] | Filter (50) | 82.2±13 | 86.6 |
| | HDG-select (Proposed) | Filter-wrapper (30) | 100±0 | 100 |

## Supporting information

**S1 Dataset. The microarray dataset of leukemia cancer in csv format.**
(CSV)

**S2 Dataset. The microarray dataset of colon cancer in csv format.**
(CSV)

**S3 Dataset. The microarray dataset of prostate cancer in csv format.**
(CSV)

**S4 Dataset. The microarray dataset of breast cancer in csv format.**
(CSV)

**S5 Dataset. The microarray dataset of central nervous system cancer in csv format.**
(CSV)

**S6 Dataset. The microarray dataset of ovarian cancer in csv format.**
(CSV)

**S1 Table. The accuracy percentage of selecting attributed genes using Filter-SVM and Filter-GBPSO-SVM approach compared to that of the original dataset.**
(DOCX)

**S2 Table. The accuracy percentage of Filter-SVM and Filter-GBPSO-SVM algorithm upon various soft files of high dimensional datasets.**
(DOCX)

**S3 Table. Comparison of the accuracy result of the proposed HDG-select application to those reported in literature.**
(DOCX)

## Author Contributions

**Conceptualization:** Shilan S. Hameed.

**Data curation:** Shilan S. Hameed.

**Formal analysis:** Shilan S. Hameed, Fahmi F. Muhammadsharif.

**Funding acquisition:** Rohayanti Hassan.

**Investigation:** Shilan S. Hameed, Liza Abdul Latiff.

**Methodology:** Shilan S. Hameed.

**Resources:** Fahmi F. Muhammadsharif.

**Software:** Shilan S. Hameed.

**Supervision:** Rohayanti Hassan, Wan Haslina Hassan, Liza Abdul Latiff.

**Validation:** Shilan S. Hameed, Fahmi F. Muhammadsharif.

**Writing – original draft:** Shilan S. Hameed.

**Writing – review & editing:** Rohayanti Hassan, Wan Haslina Hassan, Fahmi F. Muhammadsharif, Liza Abdul Latiff.

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
