## [Decision Letter · Decision Letter 0]

5 Oct 2020

PONE-D-20-11244

A novel GUI based stand-alone application using Filters-GBPSO-SVM algorithm for the selection and classification of attributed genes in high dimensional datasets

PLOS ONE

Dear Dr. Hameed,

Thank you for submitting your manuscript to PLOS ONE. After careful consideration, we feel that it has merit but does not fully meet PLOS ONE’s publication criteria as it currently stands. Therefore, we invite you to submit a revised version of the manuscript that addresses the points raised during the review process.

Reviewers identified multiple issues with the clarity of the manuscript, particularly with regard to differences of the proposed approach compared to existing approaches.  A major revision may be able to address these issues.  In particular, the revision should clearly articulate the scientific rationale for the submitted work and clearly outline how it differs from past work.

We look forward to receiving your revised manuscript.

Kind regards,

Bryan C Daniels

Academic Editor

PLOS ONE

Journal Requirements:

2. We note you have included a table to which you do not refer in the text of your manuscript. Please ensure that you refer to Table 6 in your text; if accepted, production will need this reference to link the reader to the Table.

Reviewers' comments:

Reviewer's Responses to Questions

**Comments to the Author**

1. Is the manuscript technically sound, and do the data support the conclusions?

Reviewer #1: Yes

Reviewer #2: Partly

2. Has the statistical analysis been performed appropriately and rigorously? 

Reviewer #1: Yes

Reviewer #2: Yes

3. Have the authors made all data underlying the findings in their manuscript fully available?

Reviewer #1: Yes

Reviewer #2: Yes

4. Is the manuscript presented in an intelligible fashion and written in standard English?

Reviewer #1: Yes

Reviewer #2: No

5. Review Comments to the Author

Reviewer #1: Dataset reduction == feature engineering/selection

"The SVM classifier assesses the performance of each candidate subset using 10 folds cross-validation" type: should be 10 fold cross validation.

Check grammar on this sentence: "The software application was established by means of interfacing Weka with MATLAB features using graphical user interface (GUI) in MATLAB. It"

Check grammar: "which is usually used by data analytics tool."

Check grammar: "A novel stand-alone application was successfully established"

For Figs 9 and 10, please change color scheme to be colorblind friendly (not red/green).

Where is Table 5? Jumps from 4 to 6.

Tables 3,4,6 would be easier to interpret as figures. If this is changed, please include the tables as supplemental information.

Reviewer #2: The language is unclear and needs to be checked as some parts are difficult to follow or even understand.

Moreover, in the introduction I do not clearly understand why your algorithm is a novelty in comparison to the approach currently used in literature and why is so different. I suggest to reorganise and describe in a clearer way the currently used approach, and then state the differences with the proposed approach. In particular, the part from line 84 to 103 was not clear to me. Also, it is not clear why you decided to couple GBPSO to SVM and what is the advantage; if GBPSO select the genes, why you have to use a machine learning model to classify the genes already selected?

In line 114-115 there is a repetition of the concept already stated in 111-112. Moreover, you developed a tool which seems to have no name (or it is not clearly stated). You frequently call it "application" or similar term. I think that the manuscript will be clearer by giving a name to the tool and use it to call it in the text. Also, as it has no name, I was unable to ascertain if the link present in the text points to the gitHub page to download it. If it is, then you should provide in the gitHub page instruction for the installation, a manual and an example database that can be used for test.

The figure 1 and 2 can be improved to be clearer and easier to read: why the boxes have different shapes? why are they in purple? You may try to reduce the text in the flows to explain more in the captions, use explicative shapes and color to help the reader.

The information in table 1 and 2 are repetitions of the information given in the text from line 150 to 204. I think that this part may be improved to be more fluent and not a big list of data. Maybe maintaining the tables and removing the most numbers from the written part will improve the clearness. Also, some of the descriptions of the study where the database are taken may be too detailed, giving to the reader details that are not needed to understand the proposed algorithm. I suggest to lighten this part to help the reader and the clearness.

The statement in line 219-220 "since the mean values of gene expression are affected by the high variance" is a repetition of what already stated in line 216-217 "Those with high variance affect the mean and median values of the expression of individual genes". Moreover, in Line 218 you called a "median criterion" which has not been introduced before. I have observed in the text that you frequently use term that may be explicative to you but, as they are not introduced nor explained, cannot be clearly understood. Another example is the term "feature" introduced first in line 215 which is not clear to what it refers. To correctly introduce these terms, you may decide to use them as i.e.: "to simplify the initial selection, similar genes should be removed. To this purpose authors usually utilise two criterion: the mean and the median criterion. The first implies (...); the second (...)". Also, please clarify what is the meaning of "similar genes".

The structure of the statements in line 221-224 can be simplified to be clearer. Also, in this statement you introduce a new approach without giving any information: nor why it is new, nor some short description, nor it strong points and drawback. The reader here have to decide to believe in you and not understanding what is going on or to read the other article to understand.

The whole "Statistical filters for gene selection" paragraph needs to be fully revised. At line 239 you introduced the term weights with no description or introduction of the usage, meaning, scope.

The statement in line 250-251 is not formally correct: it is true that t-test consider equal mean and variance, but between the two groups in analysis; in the text this part is missing and the reader may get lost. Also, it is not true that the t in the t-test is equal to 0 or 1, as it can assume value from 0 to 1 (line 257).

Conversely, I think that the description of the functioning of the t-test is not important for explaining the algorithm you have developed. I consider more important to describe shortly the test, the differences between a parametric and non-parametric test and why you chose to use a non-parametric one and how you implement it in your application, adding also some literature to support your decision.

The equations 2 and 3 needs to be described more in detail, to help the reader understand their functioning and why you have decided to use them.

The paragraph "Selection and classification using a wrapper-based GBPSO-SVM algorithm" also needs to be revised: first you have to describe GBPSO, SVM, and what they do (which, in reality, should have been done previously in the introduction), and then describe why you have decided to couple them and how.

Are the figure from 3 to 6 portion of the figure 7? if is that so, please, use only the figure 7, which is much more explicative and gives an idea of the interface the user will face. Maybe you can consider to use letters in the figure 7 to help the reader follow the description of the procedure.

The text from line 400 to line 442 needs to have its own paragraph, as it is no more the description of the GUI based application, but here you describe the results of the test performed to test the you algorithm and application. First of all you have to describe what you are comparing, how you have calculated the accuracy and then presents the results. Moreover, in your paper the section "results and discussion" is missing, and it should be there that you presents the results of the tests and comment them.

Your application needs to be compared to other available tool or approach, and you need to shortly introduce and describe them in the Introduction. This part is completely absent.

What is the purpose to compare biomarkers of different studies? is there a meaning in comparing biomarkers of influenza A and breast cancer? why are you making these observations? Are they important to understand your application and its performances?

In the conclusion, you stated that "A novel stand-alone application was successfully established which can be used to select and classify the most attributed genes in high dimensional data rapidly and efficiently. The application was validated on 11 datasets and it was found to perform well on most of high dimensional datasets. " but you have provided no information about the time needed to perform the analysis with your application and the other tool used as comparison. Also, you can state that your tool perform well only in comparison to other approaches or tools, having good accuracy is not enough; consider that accuracy is not the only way to compare performance, it has its own meaning and drawbacks which you have not considered. An analysis can return 100% of accuracy but can have poor precision compared to other, for example.

Lastly, please revise your manuscript keeping in mind the structure and the meaning of each part of an article:

-introduction: here you introduce the problem and your solution by also giving to the reader all the information and notions needed to fully understand the paper. These needed to be only introduced and do not have to be fully described in detail. Also, you need to say why your solution is important

-materials and methods: here you have to describe in detail your work: the algorithm, the application, the dataset used to test it, the other tools and approaches used as a comparison, the statistics used to reassume the performance of the tools/approaches

-results: here you presents the results of the test and the comparisons done

-discussion: where you discuss the results. Results and discussion may sometimes go in a fused "results and discussion part", if it makes the paper clearer and easier to understand

-conclusion: here, and nowhere else in the paper, you make some conclusions based only on the results you have presented, no more no less. If you made tests that you have not inserted in the manuscript, you have to cite them in the text, adding that you will not show the data, and the you can discuss them and make some conclusion about them.

6. PLOS authors have the option to publish the peer review history of their article (what does this mean?). If published, this will include your full peer review and any attached files.

Reviewer #1: No

Reviewer #2: No

---

## [Author Response · Author response to Decision Letter 0]

4 Nov 2020

Subject: Response to reviewer’s comments

Manuscript Number: PONE-D-20-11244

Manuscript Title: A novel GUI based stand-alone application using Filters-GBPSO-SVM algorithm for the selection and classification of attributed genes in high dimensional datasets

Dear Professor 

Bryan C. Daniels 

Academic Editor

PLOS ONE

 We acknowledge the constructive comments and considerations received from reviewers and editor to improve the contents of our manuscript. We gratefully think that the comments have helped us to strengthen the contents of our paper. The manuscript was carefully revised, the required amendments were performed and the revised sections were highlighted throughout the manuscript. Furthermore, the manuscript title was modified to appropriately cover the presented work. 

 Please find below our response to the reviewers’ comments. We hope the revised manuscript is now acceptable for publication in your journal. 

Reviewer #1: 

- Dataset reduction == feature engineering/selection.

Response:

We meant by Dataset reduction, the dimensionality reduction of the dataset which was performed through removing the genes whose expression values are close to each other among the samples. This term was revised throughout the manuscript.

- "The SVM classifier assesses the performance of each candidate subset using 10 folds cross-validation" type: should be 10 fold cross validation.

Response:

This change has been made throughout the manuscript. 

- Check grammar on this sentence: "The software application was established by means of interfacing Weka with MATLAB features using graphical user interface (GUI) in MATLAB. It"

- Check grammar: "which is usually used by data analytics tool."

- Check grammar: "A novel stand-alone application was successfully established"

Response:

The grammar of the sentences was double checked and a careful language proofread was performed throughout the whole manuscript contents.

- For Figs 9 and 10, please change color scheme to be colorblind friendly (not red/green).

Response:

The color scheme of the Figures was changed to colorblind friendly one and further discussion was given. 

- Where is Table 5? Jumps from 4 to 6.

Response:

We apologize for the typo. Table 6 is Table 5. It was corrected in this revised version. 

- Tables 3,4,6 would be easier to interpret as figures. If this is changed, please include the tables as supplemental information.

Response:

The tables have been interpreted as figures and the tables were moved to the supplementary information.

Reviewer #2: 

- The language is unclear and needs to be checked as some parts are difficult to follow or even understand. Moreover, in the introduction I do not clearly understand why your algorithm is a novelty in comparison to the approach currently used in literature and why is so different. I suggest to reorganise and describe in a clearer way the currently used approach, and then state the differences with the proposed approach. In particular, the part from line 84 to 103 was not clear to me. Also, it is not clear why you decided to couple GBPSO to SVM and what is the advantage; if GBPSO select the genes, why you have to use a machine learning model to classify the genes already selected? 

Response:

A careful language proofread was performed on the whole manuscript. The novelty of the proposed application and the difference between the current algorithm and those reported in literature are also given in the introduction section. Taking these comments into consideration, the introduction section has been revised accordingly. These changes can be found in Line 82-141. 

- In line 114-115 there is a repetition of the concept already stated in 111-112. Moreover, you developed a tool which seems to have no name (or it is not clearly stated). You frequently call it "application" or similar term. I think that the manuscript will be clearer by giving a name to the tool and use it to call it in the text. Also, as it has no name, I was unable to ascertain if the link present in the text points to the gitHub page to download it. If it is, then you should provide in the gitHub page instruction for the installation, a manual and an example database that can be used for test.

Response:

The repetitive concepts were merged and the tool was given the name HDG-select, which is derived from high dimensional gene selection. A GitHub link for the app is provided with manual and dataset for testing. 

- The figure 1 and 2 can be improved to be clearer and easier to read: why the boxes have different shapes? why are they in purple? You may try to reduce the text in the flows to explain more in the captions, use explicative shapes and color to help the reader. The information in table 1 and 2 are repetitions of the information given in the text from line 150 to 204. I think that this part may be improved to be more fluent and not a big list of data. Maybe maintaining the tables and removing the most numbers from the written part will improve the clearness. Also, some of the descriptions of the study where the database are taken may be too detailed, giving to the reader details that are not needed to understand the proposed algorithm. I suggest to lighten this part to help the reader and the clearness.

Response:

Figure 1 and 2 have been revised accordingly, their color changed to black and contents modified to better understand by the readers. Based on the standards of flowchart, the start and end shape boxes are oval, while the boxes contain input or output commands are parallelogram and those include processing commands are in rectangular shape. The tables were maintained and the numbers from the written part were removed to improve the clearness. The text in the flows reduced to explain more in the captions.

- The statement in line 219-220 "since the mean values of gene expression are affected by the high variance" is a repetition of what already stated in line 216-217 "Those with high variance affect the mean and median values of the expression of individual genes". Moreover, in Line 218 you called a "median criterion" which has not been introduced before. I have observed in the text that you frequently use term that may be explicative to you but, as they are not introduced nor explained, cannot be clearly understood. Another example is the term "feature" introduced first in line 215 which is not clear to what it refers. To correctly introduce these terms, you may decide to use them as i.e.: "to simplify the initial selection, similar genes should be removed. To this purpose authors usually utilise two criterion: the mean and the median criterion. The first implies (...); the second (...)". Also, please clarify what is the meaning of "similar genes".

Response:

The repetitive statements were merged in the revised version. The term feature is representing the gene; hence it was replaced by gene throughout the manuscript. Details on mean and median criterion along with other required revisions are given in the section of dimensionality reduction using mean and median ratio. 

- The structure of the statements in line 221-224 can be simplified to be clearer. Also, in this statement you introduce a new approach without giving any information: nor why it is new, nor some short description, nor it strong points and drawback. The reader here have to decide to believe in you and not understanding what is going on or to read the other article to understand.

Response:

The statements in line 221-224 have been revised accordingly and detailed explanation was given in the section of dimensionality reduction using mean and median ratio

- The whole "Statistical filters for gene selection" paragraph needs to be fully revised. At line 239 you introduced the term weights with no description or introduction of the usage, meaning, scope. The statement in line 250-251 is not formally correct: it is true that t-test consider equal mean and variance, but between the two groups in analysis; in the text this part is missing and the reader may get lost. Also, it is not true that the t in the t-test is equal to 0 or 1, as it can assume value from 0 to 1 (line 257). Conversely, I think that the description of the functioning of the t-test is not important for explaining the algorithm you have developed. I consider more important to describe shortly the test, the differences between a parametric and non-parametric test and why you chose to use a non-parametric one and how you implement it in your application, adding also some literature to support your decision. The equations 2 and 3 needs to be described more in detail, to help the reader understand their functioning and why you have decided to use them.

Response:

The description of t-test and its application has been revised. The statement in line 250-251 has been also corrected. The difference between parametric and non-paramteric filters was given and the reason of applying the non-parametric version of was presented in the revised section of gene selection using statistical filters. 

- The paragraph "Selection and classification using a wrapper-based GBPSO-SVM algorithm" also needs to be revised: first you have to describe GBPSO, SVM, and what they do (which, in reality, should have been done previously in the introduction), and then describe why you have decided to couple them and how.

Response:

The paragraph "Selection and classification using a wrapper-based GBPSO-SVM algorithm" was changed to “genes selection using GBPSO-SVM algorithm”. Both GBPSO and SVM algorithms were described in the introduction section and the reasons of why SVM has chosen was also given. 

- Are the figure from 3 to 6 portion of the figure 7? if is that so, please, use only the figure 7, which is much more explicative and gives an idea of the interface the user will face. Maybe you can consider to use letters in the figure 7 to help the reader follow the description of the procedure.

Response:

Yes, they were part of Figure 7. So, we only use one main interface figure with complete descriptions in the revised paper. 

- The text from line 400 to line 442 needs to have its own paragraph, as it is no more the description of the GUI based application, but here you describe the results of the test performed to test the you algorithm and application. First of all you have to describe what you are comparing, how you have calculated the accuracy and then presents the results. Moreover, in your paper the section "results and discussion" is missing, and it should be there that you presents the results of the tests and comment them.

Response:

The text from line 400 to line 442 were included into the results and discussion section. The equations used to determine the accuracy and precision of the HDG-select were also given in the section of “Development of HDG-select application”. 

- Your application needs to be compared to other available tool or approach, and you need to shortly introduce and describe them in the Introduction. This part is completely absent.

Response:

A comparison of the results achieved from the HDG-select application to the pervious reported methods/tools have been made accordingly, while these methods are mentioned and cited in the introduction section. 

- What is the purpose to compare biomarkers of different studies? is there a meaning in comparing biomarkers of influenza A and breast cancer? why are you making these observations? Are they important to understand your application and its performances?

Response:

The presentation of heatmap results for two representative datasets of influenza A and breast cancer is to show the performance and effectiveness of the proposed HDG-select tool. 

- In the conclusion, you stated that "A novel stand-alone application was successfully established which can be used to select and classify the most attributed genes in high dimensional data rapidly and efficiently. The application was validated on 11 datasets and it was found to perform well on most of high dimensional datasets. " but you have provided no information about the time needed to perform the analysis with your application and the other tool used as comparison. Also, you can state that your tool perform well only in comparison to other approaches or tools, having good accuracy is not enough; consider that accuracy is not the only way to compare performance, it has its own meaning and drawbacks which you have not considered. An analysis can return 100% of accuracy but can have poor precision compared to other, for example.

Response:

In the revised version of the manuscript, the application was edited so as to determine the precision of the gene selection and classification. Also, the computational time that the proposed application is spent on the completion of gene selection and classification from the beginning to the end is given in the revised version. Furthermore, the results are compared with those of other tools or approaches reported in literature. 

Lastly, please revise your manuscript keeping in mind the structure and the meaning of each part of an article:

-introduction: here you introduce the problem and your solution by also giving to the reader all the information and notions needed to fully understand the paper. These needed to be only introduced and do not have to be fully described in detail. Also, you need to say why your solution is important

-materials and methods: here you have to describe in detail your work: the algorithm, the application, the dataset used to test it, the other tools and approaches used as a comparison, the statistics used to reassume the performance of the tools/approaches

-results: here you presents the results of the test and the comparisons done

-discussion: where you discuss the results. Results and discussion may sometimes go in a fused "results and discussion part", if it makes the paper clearer and easier to understand

-conclusion: here, and nowhere else in the paper, you make some conclusions based only on the results you have presented, no more no less. If you made tests that you have not inserted in the manuscript, you have to cite them in the text, adding that you will not show the data, and the you can discuss them and make some conclusion about them.

Response:

Thank you for the constructive comments and the guidance on each section. We have followed the necessary corrections and modifications in order to strengthen the contents of our paper.

The revised manuscript is submitted for your positive consideration and we hope that the revised manuscript is now acceptable for publication.

Yours sincerely

Shilan S. Hameed

On-behalf of all the co-authors 

shilansamin@gmail.com

---

## [Decision Letter · Decision Letter 1]

13 Jan 2021

HDG-select: A novel GUI based application for gene selection and classification in high dimensional datasets

PONE-D-20-11244R1

Dear Dr. Hameed,

We’re pleased to inform you that your manuscript has been judged scientifically suitable for publication and will be formally accepted for publication once it meets all outstanding technical requirements.

Kind regards,

Bryan C Daniels

Academic Editor

PLOS ONE

Additional Editor Comments (optional):

One reviewer suggests additional references that you may decide to include.

Reviewers' comments:

Reviewer's Responses to Questions

**Comments to the Author**

1. If the authors have adequately addressed your comments raised in a previous round of review and you feel that this manuscript is now acceptable for publication, you may indicate that here to bypass the “Comments to the Author” section, enter your conflict of interest statement in the “Confidential to Editor” section, and submit your "Accept" recommendation.

Reviewer #3: All comments have been addressed

Reviewer #4: (No Response)

2. Is the manuscript technically sound, and do the data support the conclusions?

Reviewer #3: Yes

Reviewer #4: Partly

3. Has the statistical analysis been performed appropriately and rigorously? 

Reviewer #3: Yes

Reviewer #4: Yes

4. Have the authors made all data underlying the findings in their manuscript fully available?

Reviewer #3: Yes

Reviewer #4: (No Response)

5. Is the manuscript presented in an intelligible fashion and written in standard English?

Reviewer #3: Yes

Reviewer #4: Yes

6. Review Comments to the Author

Reviewer #3: Non.......................................................................................................................................................................................................................................................................................................................................................................................

Reviewer #4: 1. The selection and classification of genes is essential for the identification of related genes to a specific disease. Developing a user-friendly application with combined statistical rigor and machine learning functionality to help the biomedical researchers and end users is of great importance. Author developed a new stand-alone application, which is based on graphical user interface (GUI) to perform the full functionality of gene selection and classification in high dimensional datasets. HDG application is validated on eleven high dimensional datasets of the format CSV and GEO soft.

2. In gene selection research, now I consider this revised contribution as good enough to make a scientific paper valuable for the field of interest. I am a strong advocate of introducing the rigor of feature selection in classification research. However, I read the manuscript and I believe the work while having significance in the introduction of HDG technique to the problem of interest, can be suitable for publication.

3. Good paper and improved revised article.

4. Include some of the latest and relevant references for the benefit of the readers/authors of evolutionary/feature selection based journal. The following citations will be very useful for the current, future and young research scholars in this research field from all over the globe.

a. Feature selection inspired by human intelligence for improving classification accuracy of cancer types.

b. Knowledge discovery in medical and biological datasets by integration of Relief-F and correlation feature selection techniques.

c. Multi-population adaptive genetic algorithm for selection of microarray biomarkers.

d. A study on metaheuristics approaches for gene selection in microarray data: algorithms, applications and open challenges

e. Hybrid approach for gene selection and classification using filter and genetic algorithm.

7. PLOS authors have the option to publish the peer review history of their article (what does this mean?). If published, this will include your full peer review and any attached files.

Reviewer #3: **Yes: **Zakariya Yahya Algamal

Reviewer #4: No

---

## [Editor Report · Acceptance letter]

18 Jan 2021

PONE-D-20-11244R1 

HDG-select: A novel GUI based application for gene selection and classification in high dimensional datasets 

Dear Dr. Hameed:

I'm pleased to inform you that your manuscript has been deemed suitable for publication in PLOS ONE. Congratulations! Your manuscript is now with our production department. 

Kind regards, 

on behalf of

Dr. Bryan C Daniels 

Academic Editor

PLOS ONE